# Metabolite Biomarkers of *Leishmania* Antimony Resistance

**DOI:** 10.3390/cells10051063

**Published:** 2021-04-30

**Authors:** Sneider Alexander Gutierrez Guarnizo, Zemfira N. Karamysheva, Elkin Galeano, Carlos E. Muskus

**Affiliations:** 1Programa de Estudio y Control de Enfermedades Tropicales, Facultad de Medicina, Universidad de Antioquia, Medellín 050010, Colombia; sneider.gutierrez@udea.edu.co; 2Department of Biological Sciences, Texas Tech University, Lubbock, TX 79409, USA; zemfira.karamysheva@ttu.edu; 3Grupo de Investigación en Sustancias Bioactivas-GISB, Universidad de Antioquia, Medellín 050010, Colombia; elkin.galeano@udea.edu.co

**Keywords:** leishmaniasis, metabolome, proton nuclear magnetic resonance spectroscopy (^1^H-NMR), antimony, biomarkers of resistance level, oxidative stress balance, energy metabolism

## Abstract

*Leishmania* parasites cause leishmaniasis, one of the most epidemiologically important neglected tropical diseases. *Leishmania* exhibits a high ability of developing drug resistance, and drug resistance is one of the main threats to public health, as it is associated with increased incidence, mortality, and healthcare costs. The antimonial drug is the main historically implemented drug for leishmaniasis. Nevertheless, even though antimony resistance has been widely documented, the mechanisms involved are not completely understood. In this study, we aimed to identify potential metabolite biomarkers of antimony resistance that could improve leishmaniasis treatment. Here, using *L. tropica* promastigotes as the biological model, we showed that the level of response to antimony can be potentially predicted using ^1^H-NMR-based metabolomic profiling. Antimony-resistant parasites exhibited differences in metabolite composition at the intracellular and extracellular levels, suggesting that a metabolic remodeling is required to combat the drug. Simple and time-saving exometabolomic analysis can be efficiently used for the differentiation of sensitive and resistant parasites. Our findings suggest that changes in metabolite composition are associated with an optimized response to the osmotic/oxidative stress and a rearrangement of carbon-energy metabolism. The activation of energy metabolism can be linked to the high energy requirement during the antioxidant stress response. We also found that metabolites such as proline and lactate change linearly with the level of resistance to antimony, showing a close relationship with the parasite’s efficiency of drug resistance. A list of potential metabolite biomarkers is described and discussed.

## 1. Introduction

Leishmaniasis is a group of vector-borne tropical diseases considered to be one of the most epidemiologically important as it affects approximately 2 million people per year [1,2]. Leishmaniasis is caused by *Leishmania* parasites, a genus of kinetoplastids that exhibit two different stages during their life cycle: the promastigote stage adapted to survive in the gut of phlebotomine sandfly vectors, and the amastigote stage, evolutionarily adapted to survive inside the vertebrate host’s macrophages [3].

Once *Leishmania* is established in the vertebrate host, the clinical outcome is the result of the integration of multiple factors including the *Leishmania* species and/or strain, the host’s immune response, and the presence of coinfections. Consequently, the disease can be present as a wide spectrum of clinical outcomes grouped in four clinical forms: simple or diffuse cutaneous leishmaniasis (CL) characterized by local or diffuse skin ulcers; mucocutaneous leishmaniasis (ML) affecting mainly nasal mucosa; visceral leishmaniasis (VL), a systemic infection affecting organs containing macrophages such as the bone marrow, liver, spleen, and lymph nodes; and Post-kala-azar dermal leishmaniasis (PKDL) considered a dermal sequela of VL. Notably, the VL can be lethal in the absence of treatment [4].

For the last seven decades, pentavalent antimonial (Sb^V^) has been used as the primary leishmaniasis therapy. It is commonly accepted that Sb^V^ acts as a prodrug, requiring reduction to the active trivalent antimony (Sb^III^). Evidence shows that Sb^III^ induces oxidative and osmotic stress [5,6], inhibits the glycolytic pathway and fatty acid β-oxidation [7], interferes with the purine salvage pathway [8], inhibits DNA topoisomerase I [9], and competes with zinc (Zn^II^) for its binding to the CCHC and CCCH zinc finger domains [10].

Unfortunately, *Leishmania* has developed resistance to antimonials, leading to over sixty percent inefficacy in Bihar, India [11], a phenomenon that has also been seen in North Africa and Latin America [12,13,14,15,16,17]. Treatment failures due to drug resistant organisms are very common for *L. donovani, L. braziliensis/L. guyanensis, L. tropica,* and *L. major,* species responsible for visceral and cutaneous leishmaniasis. Hence, it is important for the clinician to know the isolate’s drug resistance status in order to make the correct therapeutic choice. Other drugs such as paromomycin, amphotericin b, and miltefosine, have been employed, but *Leishmania* has also shown clinical resistance against all of them [18,19,20]. Furthermore, there is an evidence of cross-resistance between antimonial and other drugs such as amphotericin b and paromomycin [19,21]. As a result, antimonials are still the main therapeutic option in several endemic countries including Colombia, Venezuela, and Brazil [22]. Together, these facts suggest that drug resistance can be considered one of the main challenges to efficiently combat leishmaniasis, and antimony resistance is a phenomenon with high clinical impact that needs to be further studied. However, no molecular methods are currently validated to track drug-resistant isolates in clinical setting. Therefore, there is an urgent need to identify biomarkers suitable to use in clinical settings and develop standardized molecular methods to measure drug resistance in clinical isolates [23].

“Omics” technologies constitute a powerful approach to analyze the molecular remodeling involved in different biological processes including drug resistance by offering the opportunity to study massively, qualitatively, and quantitatively different types of biomolecules such as genes, transcripts, proteins, lipids, or metabolites at a global cell level [24].

Studies of resistance to antimony using omics technologies have mostly focused on the genomic, transcriptomic, and proteomic levels [25,26]. Thus far, the evidence suggests that Sb^III^-resistant parasites combat antimony in several main ways: (1) decreasing drug uptake by downregulation of the aquaglyceroporin (AQP1) transporter [27]; (2) inhibiting drug by formation of inactive thiol–Sb^III^ complex and activating the thiol metabolism leading to the production of trypanothione, an important molecule involved in antioxidant response [28]; (3) increasing drug efflux by overexpression of ABC transporters that transport the thiol–metal complex toward vesicles that are secreted via exocytosis [29,30,31]; and finally (4) remodeling of carbon and lipid metabolism, which has been consistently documented in Sb^III^-resistant parasites, probably to optimize energy metabolism and modify the cell membrane composition [26,32,33,34].

However, few studies address the metabolome of *Leishmania’s* resistant phenotypes. Metabolomics is one of the latest omics technologies that has been applied successfully in many areas of life sciences, being particularly useful for phenotypic analysis since metabolites are the endpoints of the active pathways and therefore they are optimal indicators of the cell’s physiology [35]. Because of this, metabolites could potentially be used as biomarkers to differentiate antimony-resistant parasites.

Furthermore, studies have mainly focused on intracellular metabolism rather than changes at the extracellular level [36,37,38]. Interestingly, the exometabolome or metabolic footprinting is not only informative of the metabolic changes in response to different environmental conditions but also it requires an easier sample preparation than intracellular metabolomic analysis, offering the opportunity to simplify the phenotypic characterization [39].

Mass spectrometry analysis (MS) has shown that antimony-sensitive and -resistant clinical isolates exhibit dramatic differences in metabolomic profiles, particularly in lipid metabolism [40], sulfur-containing amino acids, and polyamine biosynthetic pathways [36]. ^13^C-isotope-labeling and MS confirmed that the amino acid metabolism is remodeled by involving activation of the redox system in antimony-resistant parasites [37].

However, due to *Leishmania’s* complexity, more studies are required to identify robust and reliable biomarkers of Sb^III^ resistance. For instance, it remains unexplored as to whether the abundance of differentially expressed metabolites change proportionally to the level of Sb^III^ resistance. Indeed, these types of compounds could be more closely related with the response required to combat the drug and more informative in predicting the level of resistance to Sb^III^.

Additionally, metabolomics is based on two analytical methods, MS and nuclear magnetic resonance (NMR), which are considered complementary techniques with different advantages and disadvantages; yet, in the *Leishmania* antimony resistance field, most of the studies have used MS. Although ^1^H-NMR is less sensitive, it is more robust and reproducible, the sample preparation can be easier, and the sample is not lost during the spectra acquisition, making this a valuable technique for the search of biomarkers [41,42].

This study aimed to identify the metabolites that can be potentially used as biomarkers of the Sb^III^ resistance level across ^1^H-NMR analyses from both intracellular and extracellular extracts of *Leishmania* promastigotes.

## 2. Materials and Methods

### 2.1. Reagents

Phosphate buffered saline (Santa Cruz Biotechnology, Dallas, TX, USA, product: 362182), Schneider insect medium (Sigma Aldrich, St. Louis, MI, USA, product: S0146), potassium antimony (III) tartrate trihydrate (Sigma Aldrich, St. Louis, MI, USA, product: 28300-74-5), fetal bovine serum (Eurobio Scientific, Paris, France, product: CVFSVF0001), penicillin and streptomycin (Thermo Fischer Scientific, Waltham, MA, USA, product: 15140122), deuterium oxide (Merck, Darmstadt, Hesse, Germany, product: 7789-20-0), sodium azide (NaN_3_) (Merck, Darmstadt, Hesse, Germany, product: 26628-22-8), 3-(trimethylsilyl) propionic-2,2,3,3-d4 acid sodium salt or TSP (Sigma Aldrich, St. Louis, MI, USA, product: 24493-21-8), and 2′,7′-dichlorodihydrofluorescein diacetate or H2DCFDA (Thermo Fischer Scientific, Waltham, MA, USA, product: D399).

### 2.2. Parasites, Culture, and Drug Treatment

Three levels of Sb^III^-resistance were considered for the metabolomic analysis by ^1^H-NMR. The *Leishmania tropica* wild type or antimony sensitive strain (WT), moderately resistant (MR) isogenic derivative of WT, and highly resistant isogenic derivative (HR) were used in this study. The WT, MR, and HR strains have a growing half-maximal effective concentration (EC_50_) to Sb^III^ of 10.4 ± 0.6, 377 ± 34.93, and 631.7 ± 73.7 µg/mL, respectively, and thus they were chosen to represent a low, moderate, and high level of Sb^III^ resistance, respectively.

Both sensitive and resistant strains were independently grown in culture medium. In each case, the promastigotes were seeded at 1 × 10^6^ promastigotes/mL in a final volume of 10 mL of Schneider insect medium supplemented with 10% of fetal bovine serum (FBS), 10 units/mL penicillin, and 0.1 mg/mL of streptomycin. The MR and HR strains were treated with the drug at 5 and 10 times the estimated EC_50_ value for WT strain (EC_50_ for WT strain = 10 µg/mL Sb^III^), respectively. These drug concentrations were used since they correspond to the drug concentration used to select each strain during the drug resistance stepwise selection. The WT strain was grown in drug-free medium.

Growth was monitored by evaluating cell count using a Neubauer chamber to compare the growth phases per strain in order to synchronize the cell cultures and to avoid technical variability.

### 2.3. Extraction of Intracellular Metabolites

During the stationary phase, the promastigotes were diluted at 15–20 × 10^6^ parasites/mL using Schneider insect medium without fetal bovine serum. To isolate truly resistant parasite population and avoid leakage of metabolites, we enriched the population of live parasites or parasites without membrane damage by centrifugation of 3 mL of Ficoll-Histopaque^®^-1077 and 6 mL of diluted parasite’s solution in 15 mL conical tubes at 1800× *g* at room temperature for 10 min. After Ficoll gradient centrifugation, the parasite population without membrane damage was enriched between middle and upper phase of Ficoll gradient, and 2 mL of these phases were carefully recovered in a 15 mL conical tube (Appendix A).

The parasite’s metabolism was quenched by incubating the tube containing the recovered parasites in −80 °C ethanol until they reached 4 °C (≈10 s). After that, any manipulation was performed at 4 °C to avoid sample degradation [43].

The quenched parasites were harvested and washed twice in 1 mL of cooled PBS-1X at 1800× *g*, 4 °C for 5 min. The number of parasites between different samples was adjusted to a total of 1 × 10^7^ promastigotes, and the final pellet was obtained by centrifugation at 1800× *g*, 4 °C for 5 min, and immediately resuspended in 200 µL of methanol (−20 °C) for metabolite extraction. To homogeneously release the intracellular metabolites, we treated the parasites with 5 cycles of 1 min freezing in liquid nitrogen and thawed them at room temperature with 20 s of 3400 rpm vortex (Labnet vortex mixer 120 V).

The tubes containing the lysate were shaken overnight at 250 rpm and 4 °C. Then, the samples were incubated at −80 °C for 20 min and centrifuged at 20,000× *g*, 4 °C, for 30 min to precipitate proteins. Finally, the samples were dried using a SpeedVac concentrator (Thermo Fisher Scientific, Waltham, MA, USA, SPD111V P1) and used for ^1^H-NMR analysis [44].

### 2.4. Extraction of Extracellular Metabolites

During the stationary phase, 5 mL of medium was collected and filtered using a 0.22 µm acrodisc syringe filter, previously washed with PBS (10 times) to remove membrane additives. The collected medium was mixed with 2 parts of methanol and incubated overnight at −80 °C for protein precipitation. Then, the samples were centrifuged at 1600× *g* for 30 min, and each supernatant was recovered for lyophilization using a VirTis Bench Top 4K SP scientific followed by ^1^H-NMR analysis [44].

### 2.5. Proton Nuclear Magnetic Resonance Spectroscopy ^1^H-NMR Analysis

The dried samples were reconstituted in 550 μL of deuterated phosphate buffer (50 mM NaH_2_PO_4_, 0.02% NaN3, and 0.04 mM or 1.5 mM TSP for intracellular or extracellular extracts, respectively, pH = 7.0 ± 0.04, and D_2_O).

After this the samples were vortexed for 30 s and centrifuged at ≈16,000× *g* at 4 °C for 15 min to remove any insoluble material. Supernatants were then transferred to NMR tubes with 5 mm of diameter and 7 inches of length (14219-032, VWR).

High resolution one-dimensional ^1^H-NMR spectra were obtained on a 600 MHz Bruker Avance III spectrometer (Bruker BioSpin Ltd., Billerica, MA, USA), coupled to a 5 mm Prodigy^®^ TCI cryoprobe. Applying a standard Bruker 1D spectroscopy pre-saturation pulse sequence (noesypr1d) with optimal water suppression and 256 and 128 scans for intracellular and extracellular metabolome analysis, respectively [45]. D_2_O was used for the internal lock signal and TSP-d4 as the internal standard with a chemical shift of δ 0.0 at 300 K. All raw ^1^H-NMR spectra are available under request. The samples were randomized before analysis to avoid progressive bias.

### 2.6. Data Processing

The ^1^H-NMR data processing was adapted from Mickiewicz et al. (2019) [46]. The spectra were processed in 2 different ways. First, the spectra were annotated to reveal the type of metabolites that can be identified at the intracellular and extracellular level in *L. tropica*’s methanolic extracts using Chenomx software 8.5 (www.chenomx.com, accessed on 23 April 2021, Edmonton, AB, Canada) in combination with the public database, human metabolite database (HMDB, www.hmdb.ca; accessed on 13 March 2021).

Then, the spectra were processed in parallel using the software MestReNova 14.2. This included the referencing to the TSP peak at 0.0 ppm, apodization along t1 with an exponential of 0.5 Hz, phasing by Metabonomics algorithm, and baseline correction applying the Whittaker Smoother algorithm. The phasing and baseline quality were verified manually. Finally, the spectra were binned using the method average sum with a width of each integral region of 0.04 ppm, from 0 to 10 ppm. Bins covering signals that are not of interest, such as water and residual methanol signals, were manually removed before statistical analysis.

### 2.7. Statistical Analysis

The datasets generated after spectra binning were normalized by probabilistic quotient normalization (PQN) [47] and imported to SIMCA Umetrics V14.1 software (Umetrics, Umeå, Sweden) for multivariate statistical analysis, adapting the strategy used by Mickiewicz et al. (2018) [48]. First, the datasets were scaled by the Pareto method [49]. Then, principal component analysis (PCA) and orthogonal partial least squares discriminant analysis (OPLS-DA) were completed as unsupervised and supervised multivariate methods, respectively.

The PCA was carried out to summarize the source of variation in each dataset and explore the sample grouping and the presence of outliers. The OPLS-DA method was used to extract maximum information on discriminant signals (compounds) from the spectra.

The OPLS-DA models were validated by sevenfold cross-validation (CV), calculating the R^2^Y (the percentage of variation explained by the model), Q^2^ (the predictive ability of the model), and CV-ANOVA (cross-validated analysis of variance) *p*-value with a cutoff of 0.05 [50,51]. The significant metabolites were selected from the OPLS-DA regression coefficients and variable influence on projection (VIP) scores higher than 1 using the Jackknife technique as bias estimator (*p*-value ≤ 0.05).

S-line diagrams were performed to summarize the contribution of each VIP to the OPLS-DA models. S-line plot visualizes the signals (chemical shift) distributed by p(ctr) [1] loading and colored according to the absolute value of the correlation loading, p(corr) [1], or modeled correlation. The top end of the color scale visualizes the NMR shifts that influence the separation of the groups [52].

To identify metabolites linearly affected by the antimony resistance level, which can be more precise and reliable for the prediction of resistant phenotypes, we calculated Pearson correlation scores and filtered them by false discovered rate (FDR ≤ 0.05). The metabolites with a higher absolute Pearson coefficient were prioritized and analyzed using an open source receiver operating curve characteristics (ROC) analysis tool in MetaboAnalyst 5.0 [53].

### 2.8. Measurement of Reactive Oxygen Species (ROS) Levels

To measure the ROS production in antimony-sensitive and -resistant parasites, we used hydrogen peroxide (H_2_O_2_) as an oxidative stress inducer, adapting the methodology used by Karampetsou et al. (2019) [54]. Parasites were harvested by centrifugation at 1800× *g* for 5 min and washed twice in 1 mL of PBS 1X. Then, the cell density was normalized at 1 × 10^6^ parasites/mL. In each case, a group of parasites was incubated in PBS without any additional treatment (control group). On the other hand, an additional group was treated with 0.5 mM hydrogen peroxide (H_2_O_2_) for 4 h in a final volume of 1 mL. Once the period of exposure was completed, the pelleted parasites were resuspended in 100 µL of H_2_DCFDA at 0.4 µg/mL. H_2_DCFDA is an indicator for reactive oxygen species (ROS) production in live cells. The samples were incubated for 1 h at room temperature in darkness. Then, the parasites were washed in 1 mL of PBS and resuspended in 300 µL of fresh PBS. Staining with propidium iodide (0.125 µg/mL) was used as a cell viability control. Finally, the fluorescence of PI and H_2_DCFDA were acquired using 610/20 and 530/30 as detection filters, respectively, in a LSRFortessa™ Becton Dickinson cell flow cytometer.

## 3. Results

### 3.1. Experimental Design

Several reports demonstrated that the most significant changes in the metabolite’s composition occur during stationary phase [40,55]. Therefore, we collected samples from this stage of parasite growth. To detect metabolomic changes in Sb^III^-resistant parasites, we analyzed both intracellular and extracellular extracts of *Leishmania*’s promastigote in at least three independent biological replicates. The intracellular metabolome reflects the cell physiology, while the extracellular metabolome represents the metabolites uptake and excretion in a particular condition.

In this study, we analyzed three strains (WT, MR, and HR) representing increasing Sb^III^ resistance levels because of a progressive stepwise resistance selection in vitro. Since the strains show different phenotypes (Sb^III^ resistance levels), they were considered independent for statistical purposes.

The two resistant strains (MR and HR) were treated with Sb^III^ to stimulate their acquired mechanism of resistance, while the WT was grown without drug treatment and used as a control since the sensitive parasite cannot grow until stationary phase in the presence of the drug. Under the described experimental conditions, all strains showed a similar growth curve, eliminating growth stage bias and allowing for the comparison of ^1^H-NMR profiling between groups (Appendix A).

Typically, a ^1^H-NMR spectrum shows the proton signals distribution based on their chemical shift (a proton with a particular chemical environment) and their intensity (abundance), offering the opportunity to characterize the type of compound (fingerprint or signals pattern) and estimate its concentration (signal intensity). On this basis, the workflow represented in Figure 1 included sample preparation followed by ^1^H-HMR analysis. Spectra annotation was performed assigning the corresponding compound to each detected signal in order to define the *Leishmania*’s metabolome coverage detected by ^1^H-NMR. After spectra binning, the multivariable analysis was used to identify the group of signals (bins) significantly differentiating Sb^III^-resistant phenotypes or VIPs. The signals detected as VIPs were matched with the previously annotated spectra, and the respective compounds were suggested as potential biomarkers of Sb^III^ resistance phenotypes, prioritizing those that exhibited a significant linear correlation between metabolite concentration (signal intensity) and Sb^III^ resistance level. Finally, we interactively analyzed the intracellular and extracellular metabolome to discuss the metabolic pathways potentially affected (Figure 1).

### 3.2. Metabolomic Coverage in Intracellular and Extracellular Methanolic Extracts Using ^1^H-NMR

After ^1^H-NMR analysis, both intracellular and extracellular spectra were manually annotated to define the metabolome coverage. Usually, the metabolome coverage is limited by several factors including the metabolite concentration and the solvent. In our study here, we employed a methanol solvent for extraction since it is recognized as having a better metabolite coverage [56].

Excluding the reference compound and the solvent’s signals, we annotated 26 and 29 metabolites in the spectra from intracellular and extracellular extracts, respectively. The same compounds were detected in both Sb^III^-sensitive and -resistant parasites (Figure 2 and Figure 3, Appendix A).

Considering that some metabolites are commonly detected at the intracellular and extracellular level, a total of 40 different compounds were detected by ^1^H-NMR, 15 of which were commonly detected in both intracellular and extracellular extracts. A total of 11 and 14 compounds were exclusively identified at the intracellular or extracellular levels, respectively (Appendix A).

The type of metabolites detected were mainly distributed in three metabolite classes: amino acids (total matching compounds: 18), tricarboxylic acids (total matching compounds: 3), and purines (total matching compounds: 2) (Appendix A). Additionally, these compounds were enriched significantly in six pathways of the KEGG database, including aminoacyl-tRNA biosynthesis (total hits: 16); valine, leucine, and isoleucine biosynthesis (total hits: 4); arginine biosynthesis (total hits: 4); alanine, aspartate, and glutamate metabolism (total hits: 5); glyoxylate and dicarboxylate metabolism (total hits: 5); pantothenate and CoA biosynthesis (total hits: 2); nicotinate; and nicotinamide metabolism (total hits: 3) (Appendix A).

### 3.3. ^1^H-NMR Spectra Efficiently Differentiated the Antimony-Resistant Phenotypes Both at the Intracellular and Extracellular Levels

The PCA models were based on two principal components that in total summarized 74.2% and 43.2% of the variance in the dataset for intracellular and extracellular analysis, respectively (Figure 4A and Figure 5A). PCA score plots showed that resistant strains were clustered away from the sensitive strain, suggesting that the metabolomic profiles were different in both intracellular and extracellular extracts. Additionally, no outliers were detected since all samples were distributed within the ellipse, representing Hotelling’s T-squared 95% confidence interval (Figure 4A and Figure 5A).

The OPLS-DA analysis was carried out to reveal metabolic differences between Sb^III^-sensitive and Sb^III^-resistant parasites. Models were obtained using one predictive and one orthogonal component. The OPLS-DA score scatter plots showed a clear separation between Sb^III^-resistant (MR and HR strain) and Sb^III^-sensitive phenotypes (WT) at both the intracellular and extracellular levels (Figure 4B and Figure 5B). Furthermore, the models were correctly validated, obtaining good correlation (R2Y) and predictive (Q2) scores and significant CV-ANOVA *p*-values (R2Y > 0.6, Q2 > 0.6, R2Y ≠ Q2 < 0.3, *p*-value < 0.05) (Appendix A).

Major contributing metabolites for group separation were identified after the validation of the estimators (coefficient value and VIP score) using the Jackknife method as shown in an S-line plot. A total of seven and five metabolites at the intracellular and extracellular levels, respectively, were selected as potential biomarkers of antimony resistance. At the intracellular level, Sb^III^-resistant parasites showed an upregulation of alanine, proline, arginine, and lysine, and a downregulation of sn-glycero-3-phosphocholine, betaine, and acetate (Figure 4C). Additionally, at the extracellular level, Sb^III^-resistant parasites showed lower levels of proline, valine, lactate, and threonine (Figure 5C). Some of the signals with high correlation values (color bars) are not shown because they were not significant by the Jackknife method or because the associated metabolites could not be identified by signal overlapping (Figure 4C and Figure 5C). The VIP score and the coefficient size are summarized in the Appendix A.

### 3.4. Proline and Lactate Changed Linearly with the Antimony Resistance Level

Once the potentially important metabolites differentiating Sb^III^-sensitive and Sb^III^-resistant parasites were identified from the OPLS-DA models, we then explored whether the expression of those metabolites exhibited a significant linear correlation between the Sb^III^ resistance level and the metabolite abundance. Interestingly, proline metabolite abundance showed a positive correlation with the Sb^III^ resistance level in intracellular extracts, while both proline and lactate metabolite abundance showed a negative correlation with Sb^III^ resistance level in extracellular extracts (Figure 6 and Figure 7).

To evaluate the importance of discriminating metabolites, we applied the receiver operating characteristic (ROC) curve analysis to the metabolites with a higher correlation value as a binary classifier (sensitive vs. resistant strains). ROC analyses showed optimal sensitivity and specificity scores with areas under the curve (AUCs) of 1 for all evaluated metabolites in a univariate approach (Figure 6 and Figure 7). However, these AUC scores should be carefully interpreted since here we analyzed only one *Leishmania* species using an in vitro approach, and consequently the metabolites were described as potential biomarkers.

Even though glycerol-phosphocholine and betaine did not show a significant Pearson correlation, they still showed a high capability to differentiate Sb^III^-sensitive and Sb^III^-resistant parasites on the basis of the AUC score, and thus they were also included (Figure 6).

## 4. Discussion

Antimony resistance remains an important determinant in leishmaniasis treatment, and biomarkers of resistant phenotypes can help to select an optimal medication. Here, we applied metabolomics to *L. tropica* strains with different resistance levels to Sb^III^ as biological model. Clinical isolates of *L. tropica* are known to exhibit progressive reduction of antimony susceptibility, demanding strategies to identify and prevent the dissemination of drug-resistant strains [17]. Through the multivariate statistical analysis, we were able to identify 10 different metabolites (proline, alanine, arginine, lysine, betaine, glycerol-phosphocholine, lactate, acetate, threonine, and valine) differentiating the metabolic profiles of Sb^III^-resistant and -sensitive parasites at the intracellular and/or extracellular levels (Figure 4C and Figure 5C).

Proline was the only significant metabolite differentiating antimony-sensitive and -resistant parasites in both intracellular and extracellular extracts. Specifically, a decreased proline abundance at the extracellular level and an increased proline abundance at the intracellular level were representative features of the Sb^III^-resistant phenotypes. A similar observation was reported also for another *Leishmania* species: *L. infantum* promastigotes selected for Sb^III^ resistance, and *L. donovani* promastigotes derived from clinical isolates; both of which showed higher proline concentration at the intracellular level [37,40]. Additionally, proline upregulation was also observed during both logarithmic and stationary phases of Sb^III^-resistant *Leishmania* growing under drug pressure [32]. However, these studies did not include the exometabolomic analysis.

There was also the observation that the parasite showed better tolerance to Sb^III^ in the medium with high proline concentration but not against other drugs such as amphotericin or miltefosine [57]. Remarkably, in our study, we found that proline concentration was differentially increased in Sb^III^-resistant parasites at the intracellular level. Additionally, the exometabolomic analysis confirmed that resistant parasites depleted proline from the medium at higher level. The fact that the changes of proline metabolism can be detected at the extracellular level makes the exometabolomic analysis useful for Sb^III^ resistance profiling. Interestingly, in this study, both intracellular and extracellular analysis showed that proline concentration changes linearly with the drug resistance level. This evidence indicates that proline levels are closely associated with the parasite’s ability to efficiently combat the drug and could be considered a robust metabolite biomarker of antimony resistance in *Leishmania* (Figure 6 and Figure 7).

In different types of eukaryotic cells including *Leishmania*, proline acts as an osmolyte and regulates reactive oxygen species (ROS). Is commonly accepted that antimonials induce a ROS imbalance [5,6]. Interestingly, here we also verified that the level of resistance to Sb^III^ was associated with better response to combat the oxidative stress induced in vitro by hydrogen peroxide (H_2_O_2_). This free radical is commonly produced by the host macrophages during phagocytosis, inducing promastigote death in an apoptosis-like manner [58] (Appendix A).

Notably, proline analogues have been explored as promising therapeutic alternatives in trypanosomes [59]. In *Leishmania* parasites, ω-amino acid 4-amino-butiric (GABA), a proline analogue, inhibits the proline/alanine transporter and causes toxicity in both promastigotes and amastigotes [60]. This indicates that proline transport inhibitors could potentially be used to reverse Sb^III^-resistant phenotypes. More studies are needed to evaluate the effect of combined proline analogue and antimonial therapy.

Alanine, another amino acid contributing to the osmotic balance in *Leishmania,* was also found to be significantly increased in resistant parasites [61]. Both alanine and proline have common transport in *Leishmania* and it has been hypothesized that the regulation of the proline/alanine transporters such as LdAAP24 are key to osmotic stress responses in environments such as those induced by Sb^III^ [62]. Another study has also detected alanine to be differentially expressed in sensitive parasites growing under Sb^III^ pressure [36], suggesting that the alanine increase may not be exclusive of resistant phenotypes.

At the intracellular level, betaine and glycerol-phosphocholine were significantly decreased in Sb^III^-resistant parasites at the intracellular level. Betaine is a donor of methyl groups while glycerol-phosphocholine is a precursor for choline biosynthesis [63]. Choline can feed into phosphatidylcholine biosynthesis but also can be oxidized to produce betaine [64]. There is an evidence that phosphatidylcholine biosynthesis is decreased in *Leishmania* Sb^III^-resistant parasites growing under drug challenge, supporting the idea that glycerol-phosphocholine degradation taking place in Sb^III^-resistant parasites is necessary to increase methyl donor availability [40,65].

Together, these observations suggest that the decrease detected in betaine and sn-glycero-3-phosphocholine could be associated with their ability to donate or release methyl groups. Methyl groups are highly required to feed the methionine cycle and then the polyamine biosynthesis, two pathways interconnected by the decarboxylated S-adenosyl-L-methionine. Consistently, polyamine biosynthesis and the downstream trypanothione pathway have been widely recognized to be activated in Sb^III^-resistant parasites challenged with Sb^III^ [5,66].

Other amino acids were also differentially detected. Arginine was increased in Sb^III-^resistant parasites at the intracellular level. This profile has been also detected in *L. infantum* promastigotes [37], while an arginine downregulation has been associated with sensitive parasites growing under antimony pressure [36]. It has been suggested that *Leishmania* parasites can sense arginine availability regulating the expression of *Leishmania* arginine transporter (AAP3) under stress conditions [67]. Interestingly, there is evidence that the AAP3 transporter is upregulated at the mRNA level in Sb^III^-resistant parasites derived from clinical isolates [68], which could explain the tendency toward a higher arginine intracellular concentration in Sb^III^-resistant phenotypes. Arginine is also interconnected with polyamine metabolism, producing trypanothione downstream via arginase, which catalyzes the enzymatic hydrolysis of L-arginine (L-Arg) to L-ornithine and urea [69].

We observed that lysine was increased in Sb^III^-resistant parasites at the intracellular level during the stationary phase. Conversely, other studies found that lysine was significantly increased in Sb^III^-resistant parasites during logarithmic phase [70]. It is possible that these opposing findings could be explained by the fact that the analyses were completed in different *Leishmania* species during different growth phases. However, more studies are required to clarify the role of lysine in antimony resistance.

Besides amino acids, we also selected other metabolites that could be important to differentiate the phenotypes of Sb^III^ resistance. In *Leishmania*, acetate is produced from acetyl-CoA, a product of the glycolytic pathway [71]. In trypanosomes, acetate has been shown to be an important precursor for lipid biosynthesis [72]. It is possible that the observed decline in acetate is associated with the energy and lipid production. This is supported by evidence that suggests that energy metabolism and lipid remodeling play an important role in Sb^III^ resistance [40,73].

Additionally, other metabolites involved in energy metabolism were detected. Lactate is a product of glucose catabolism. In *Leishmania*, glycolysis can take place in aerobic as well as in anaerobic conditions, and the lactate product is secreted [71]. We detected that lactate was increased in sensitive parasites at the extracellular level, suggesting that in response to Sb^III^, there is a change in energy metabolism. Consistent with this, it is commonly known that while the glycolysis is inhibited in Sb^III^-sensitive parasites, resistant parasites overexpress this pathway [26,74,75].

Furthermore, our findings suggest that resistant parasites more efficiently utilize threonine and valine since these metabolites were detected in lower concentrations in extracellular extracts. Threonine is another exogenous amino acid that can be used by *Leishmania* to supply protein synthesis and as a carbon source to feed the tricarboxylic acid cycle (TCA) [76]. Valine has been shown to be critical for *Leishmania* viability, and L-valine is degraded into pathways that lead to the production of succinyl coenzyme A, an important precursor that can be used in TCA cycle or fatty acid β-oxidation [77].

In summary, the metabolomic profiling by ^1^H-HMR allows for a clear differentiation between Sb^III^-resistant and -sensitive parasites (Figure 8). The metabolites identified in this study were associated with two general processes. The first group of metabolites included metabolites responding to osmotic and oxidative stress (proline, alanine, arginine, betaine, glycerol-phosphocholine). Some of these metabolites also contribute to the activation of thiol metabolism, which induces drug inactivation by production of the thiol–Sb^III^ complex and promotes the oxidative stress balance via trypanothione. Trypanothione system, the parasite’s equivalent to the mammalian glutathione, is the main detoxifying system against oxidative damage. Since *Leishmania* lacks catalases, trypanothione regeneration relies on nicotinamide adenine dinucleotide phosphate (NADPH) [78]. As consequence, thiol metabolism requires a large amount of energy, demanding continuous production of reductant in the form of nicotinamide adenine dinucleotide phosphate (NADPH) [79].

The second group of metabolites (acetate, threonine, valine, and lactate) can contribute to energy production via pathways such as the TCA cycle [80]. TCA cycle, malic enzyme, or pentose phosphate pathway can fuel the NADPH required for thiol metabolism activation (first group of metabolites). The coupling between the activation of thiol metabolism and energy production is essential for resistant parasites to combat the drug efficiently (Figure 8).

Currently, there are no reliable molecular methods to determine drug susceptibility in clinical settings, and treatment failure due to drug resistance is a big problem in *Leishmania* species [23]. In the absence of validated standardized methods and markers to track drug resistance in clinical isolates, the further investigation and validation of identified drug resistance biomarkers in a clinical setting is crucial. More studies are also required to validate the proposed biomarkers in other *Leishmania* species due to the high complexity of the model. Likewise, studies using gene-editing techniques are needed to clarify the metabolic adaptations under drug-induced stress conditions.

## 5. Conclusions

Sb^III^-resistant and -sensitive parasites showed significant changes in the metabolite composition and these phenotypes can be potentially predicted using ^1^H-NMR profiling. The differences in metabolite composition suggest that the development of resistance to Sb^III^ involves an optimized response to the osmotic/oxidative stress induced by the drug as well as a rearrangement of the carbon metabolism possibly to compensate the production of energy in form of reductant, typically required to keep the antioxidant response active.

At the intracellular and extracellular levels, metabolites such as proline and lactate are informative not only of the resistant phenotype but also of the level of resistance, suggesting that the metabolic pathways in which these compounds are involved are closely related with the mechanism activated to combat the drug, and consequently these metabolites can be prioritized as a more robust biomarker of Sb^III^ resistance.

In many studies, metabolomic analysis has been performed at the intracellular level; however, our study showed that simpler and time-saving exometabolomic analysis can be also efficiently used for differentiation of sensitive and resistant parasites.

## Figures and Tables

**Figure 1 cells-10-01063-f001:**
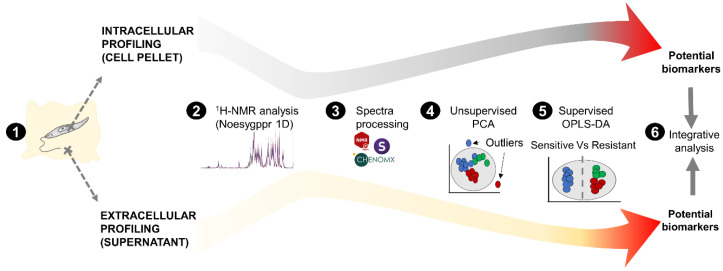
Experimental design to differentiate the metabolomic profiles in antimony-resistant and -sensitive *L. tropica* parasites. The workflow for intracellular and extracellular analysis is shown in parallel. (1) After parasite growth, the cell pellet and supernatant were collected and used for intracellular and extracellular analysis, respectively. (2) The methanol extracts were analyzed by ^1^H-HMR using a Noesygppr1d pulse. (3) Spectra processing included the annotation, apodization, alignment, phasing, baseline correction, bucketing, data normalization, and data scaling, integrating the following software: Chenomx v8.5, MestReNova v14.2, and SIMCA Umetrics v14.1. (4) Unsupervised multivariate analysis by PCA for preliminary grouping visualization and outlier detection based on the Hotelling t-squared statistic with 95% confidence limit. (5) Supervised multivariate analysis by OPLS-DA for the identification of statistically significant “variable importance in projection” (VIP) or potential biomarker, filtering by the Jackknife method to correct estimation bias. (6) Finally, the VIPs were analyzed to discuss the possible biological meaning.

**Figure 2 cells-10-01063-f002:**
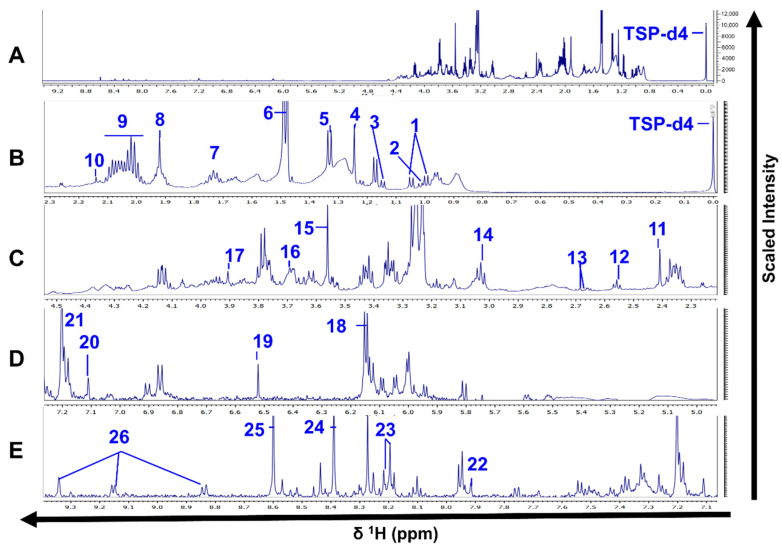
^1^H-NMR spectra of *L. tropica* intracellular extracts. (**A**) Full ^1^H-NMR spectrum from δ 0.0 to δ 9.3. (**B**) Enlarged spectrum from δ 0.0 to δ 2.3. (**C**) Enlarged spectrum from δ 2.3 to δ 4.5. (**D**) Enlarged spectrum from δ 5.0 to δ 7.2. (**E**) Enlarged spectrum from δ 7.1 to δ 9.3. Peaks of 26 compounds: 1, valine; 2, isoleucine; 3, propylene glycol; 4, 3-hydroxyisovalerate; 5, lactate; 6, alanine; 7, arginine; 8, acetate; 9, proline; 10, methionine; 11, succinate; 12, β-alanine; 13, malate; 14, lysine; 15, glycine; 16, glycerophosphocholine; 17, betaine; 18, IMP; 19, fumarate; 20, ∂-methylhistidine; 21, desaminotyrosine; 22, xanthine; 23, hypoxanthine; 24, Formate; 25, AMP; 26, NAD. TSP-d4 as an internal standard with chemical shift at δ 0.0. Only one spectrum is represented for simplification. The spectrum corresponds to the HR strain.

**Figure 3 cells-10-01063-f003:**
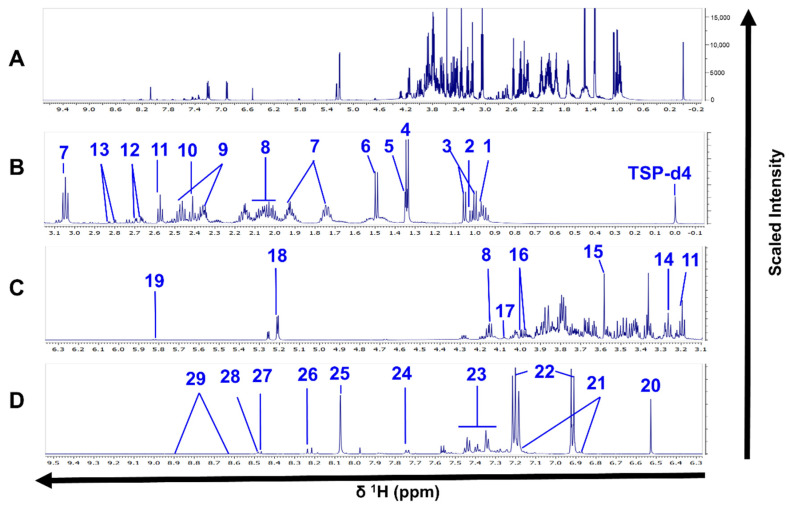
^1^H-NMR spectra of *L. tropica* extracellular extract. (**A**) Full ^1^H-NMR spectrum from *δ* 0.0 to *δ* 9.5 (TSP-*d*_4_ as an internal standard with chemical shift at *δ* 0.0). (**B**) Enlarged spectrum from *δ* 0.0 to *δ* 3.1. (**C**) Enlarged spectrum from *δ* 3.1 to *δ* 6.3. (**D**) Enlarged spectrum from *δ* 6.3 to *δ* 9.5. Peaks of 29 compounds: 1, leucine; 2, isoleucine; 2, valine; 4, lactate; 5, threonine; 6, alanine; 7, lysine; 8, proline; 9, glutamine; 10, succinate, 11, ß-alanine; 12, malate; 13, aspartate; 14, arginine; 15, glycine; 16, l-serine; 17, myoinositol; 18, trehalose; 19, uracil; 20, fumarate; 21, *N*-acetyl tyrosine; 22, tyrosine; 23, phenylalanine; 24, tryptophan, 25, methylhistidine; 26, hypoxanthine; 27, formate; 28, imidazole, 29, nicotinate. Only one spectrum is represented for simplification. The spectrum corresponds to the HR strain.

**Figure 4 cells-10-01063-f004:**
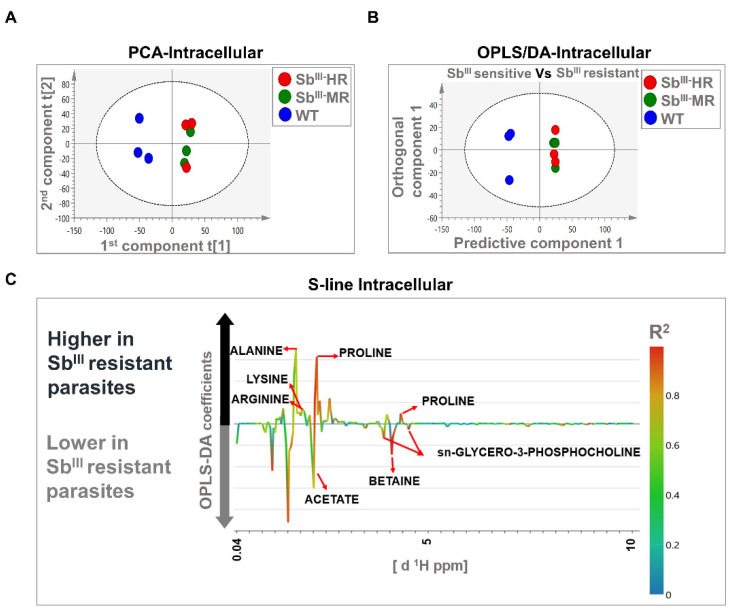
Multivariate analysis showing significant differences between the metabolomic profiles of sensitive and resistant *Leishmania* parasites at the intracellular level by ^1^H-NMR. (**A**) Score plot of PCA. (**B**) OPLS-DA score plot comparing antimony-resistant vs. antimony-sensitive parasites. (**C**) S-line plot highlighting the metabolites with major contributions to group separation. Each individual dot in panel (**A**) or (**B**) represents an observation or sample. Some dots can be overlapped. Blue dots: WT strain; green dots: MR strain; red dots: HR strain. The dotted ellipse of score plots describe the 95% confidence interval of the Hotelling’s T-squared distribution. The color bar in the S-line plot corresponds to the absolute value of the correlation loading in the discrimination model. A total of 9 observations are shown distributed in 3 independent biological replicates per experimental condition (WT, MR, and HR).

**Figure 5 cells-10-01063-f005:**
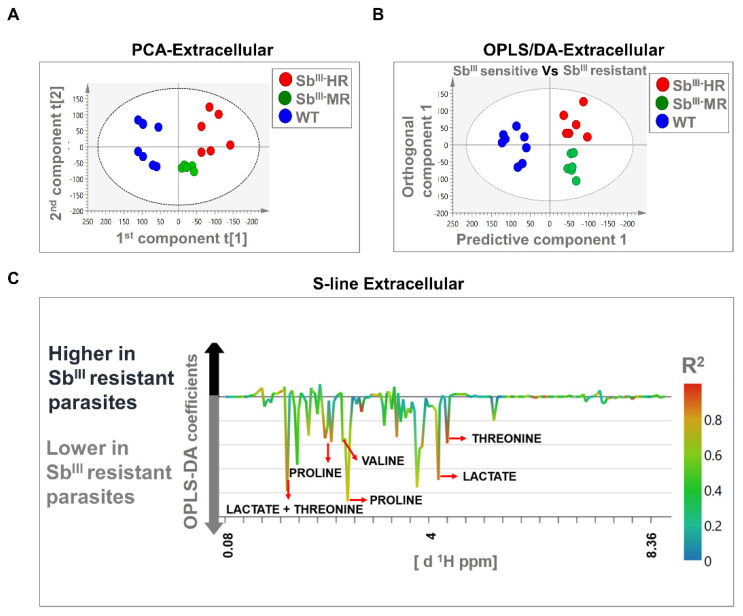
Multivariate analysis showing significant differences in metabolomic profiles of sensitive and resistant *Leishmania* parasites at the extracellular level by ^1^H-NMR. (**A**) Score plot of PCA. (**B**) OPLS-DA score plot comparing antimony-resistant vs. antimony-sensitive parasites. (**C**) S-line plot highlighting the metabolites with major contributions to group separation. Each individual dot in panel (**A**) or (**B**) represents an observation or sample. Some dots can be overlapped. Blue dots: WT strain; green dots: MR strain; red dots: HR strain. The dotted ellipse of score plots describe the 95% confidence interval of the Hotelling’s T-squared distribution. The color bar in the S-line plot corresponds to the absolute value of the correlation loading in the discrimination model. A total of 20 observations or independent biological replicates are shown, 8 of them correspond to WT, 6 to MR, and 6 to HR group.

**Figure 6 cells-10-01063-f006:**
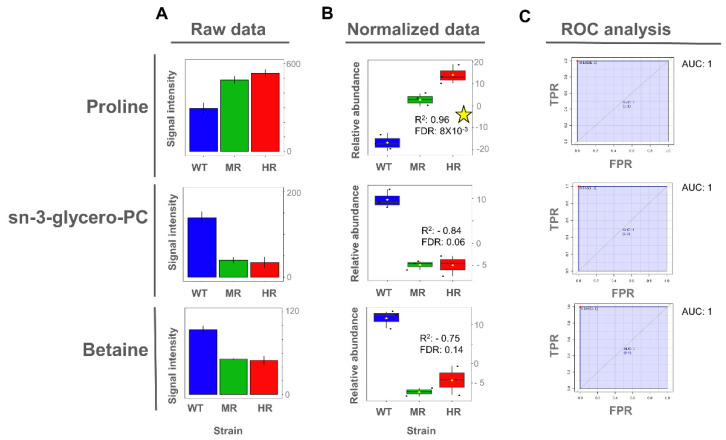
Proline, glycerol-phosphocholine, and betaine as potential biomarkers of antimony resistance at the intracellular level. (**A**) Bar plot of metabolite concentration as the raw intensity values. (**B**) Box plot of metabolite concentration as the normalized intensity values and showing the Pearson correlation scores. (**C**) Univariate ROC analysis comparing Sb^III^-resistant (MR and HR) vs. -sensitive strains (WT). Coefficient of determination by Pearson correlation (R^2^). False discovery rate of the *p*-values obtained for the R^2^ score (FDR). Significant high correlations (FDR ≤ 0.05) are highlighted with a yellow star. Sensitivity or true positive rate (TPR). Specificity or false positive rate (FPR). Area under the curve for univariate ROC analysis (AUC).

**Figure 7 cells-10-01063-f007:**
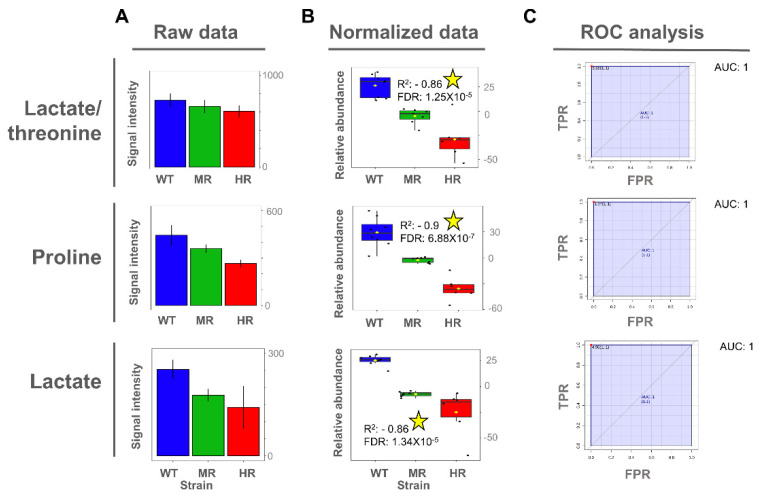
Lactate, proline, and threonine as potential biomarkers of antimony resistance at the extracellular level. (**A**) Bar plot of metabolite concentration as the raw intensity values. (**B**) Box plot of metabolite concentration as the normalized intensity values and showing the Pearson correlation scores. (**C**) Univariate ROC analysis comparing Sb^III^-resistant (MR and HR) vs. -sensitive strains (WT). Coefficient of determination by Pearson correlation (R^2^). False discovery rate of the *p*-values obtained for the R^2^ score (FDR). Significant high correlations (FDR ≤ 0.05) are highlighted with a yellow star. Sensitivity or true positive rate (TPR). Specificity or false positive rate (FPR). Area under the curve for univariate ROC analysis (AUC).

**Figure 8 cells-10-01063-f008:**
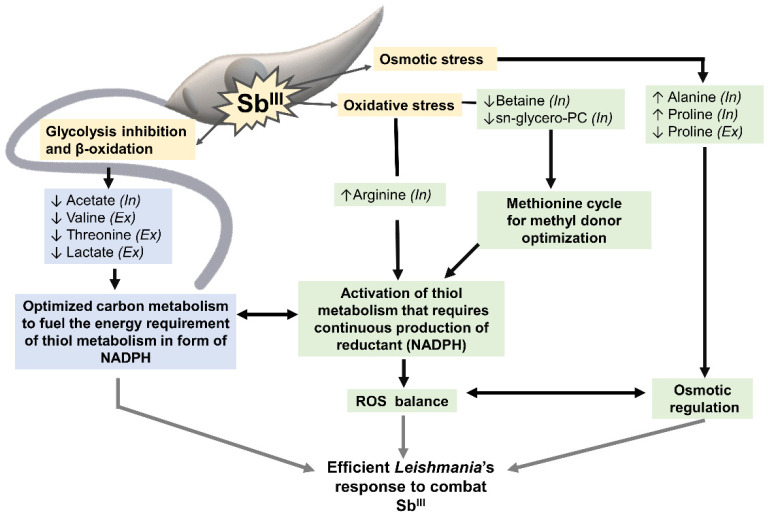
Sb^III^-resistant parasites showed a coordinated metabolomic remodeling to combat the stress caused by the drug. Yellow boxes highlight some of the effects typically caused by Sb^III^ in *Leishmania* parasites: glycolysis and β-oxidation inhibition, osmotic and oxidative stress. Green boxes highlight the group of metabolites contributing to the osmotic/oxidative balance and thiol metabolism activation. Blue boxes show the metabolites potentially contributing to the energy metabolism required to fuel thiol metabolism in the form of NADPH. This coordinated metabolic response is essential to develop resistance and combat the drug efficiently in *Leishmania* parasites. Metabolites increased in Sb^III^-resistant parasites (↑). Metabolites decreased in Sb^III^-resistant parasites (↓). Metabolites differentially detected at the intracellular *(In)* and extracellular levels *(Ex)*.

## Data Availability

The data presented in this study are available on request from the corresponding authors.

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
