# Peer review of "Metabolite Biomarkers of Leishmania Antimony Resistance"

_cells, 2021, doi:10.3390/cells10051063_

Round 1

Reviewer 1 Report

In this manuscript the authors use NMR to define metabolite differences between antimony sensitive and resistant parasites. This is a well written and clear manuscript with the data supporting their conclusions.

This work builds on previous studies of metabolism in antimony resistant parasites but uses NMR as an orthogonal approach. The authors identify some potential diagnostic metabolites and highlight how a number of metabolites scale with antimony resistance.

Overall I don't have any significant issues with this manuscript but have a number of suggestions that would improves the paper.

The authors investigate the resistance to oxidative stress in these parasites yet this data is only referred to in the discussion, which is odd. I think this is a confirmatory experiment but even so this should be integrated into the the main section of the results.

line 361-362 - in 4C I can't see the dots the legend refers to. I think the figure legend for figure 4 has sentences out of order and should be clarified so the detail for each panel is with the appropriate bit in the legend.

line 255 - delete complementary

line 404 - should be species

Author Response

Thank you for reviewing the manuscript. The changes have been introduced as requested.

Reviewer 2 Report

The paper provides evidence of metabolome modification linked to change in antimony susceptibility of Leishmania isolates. I am genuinely enthusiastic about this paper's scientific soundness, and all the comments below are aimed only at pointing to the interest of the research performed.

General comments

Please also consider PKDL and Diffuse cutaneous leishmaniases that can be relevant to cite for your paper's readers.

Since you work on L. tropica, I do not understand why you do not cite works that have depicted non-sensitive antimonial clinical isolates of L. tropica. This gives additional impact on your paper (Eddaikra et al., 2018; 10.1371/journal.pntd.0006310). To this particular point, it should be relevant to investigate further your findings on clinical isolates of leishmania falling in the framework of the AMR definition (Sereno et al., 2019 0.1016/j.actatropica.2019.01.009).

Line 56 antimony non-susceptible isolates are also identified in North Africa (Eddaikra et al., 2018; 10.1371/journal.pntd.0006310).

Do you think that such modification in the exometobolome can also be identified in the excreted body fluid of infected patients like pointed for the diagnostic of some animal trypanosomes, for example, see (Sereno et al., 10.3390/ijms21051684)?

Line 100 authors highlight the genetic diversity of Leishmania isolate and the complexity in addressing biomarker's definition to diagnose it. I firmly agree with this point discussed in this paper (Sereno et al., 2019 0.1016/j.actatropica.2019.01.009)—some proposition of definition and way to address Leishmania isolates' classification in the clinical setting. Drug resistance and therapeutic failure are not synonymous. Therapeutic failure encompasses an ensemble of factors linked to the host (i.e., Genetic, Immunologic…) to the infective agent (i.e., Drug resistance…), to the drugs (i.e., pharmacodynamics/pharmacokinetic…) and, to the chemotherapeutic protocol. Nevertheless, the first indication that informs a clinician on the therapeutic choice is the isolate's drug resistance status. Leishmania drug resistance threatens the prevention and treatment of these infections. A small number of publications deal with field/clinical identification of drug-resistant Leishmania. Treatment failure due to drug-resistant organisms is increasingly reported for L. donovani, L. braziliensis/L. guyanensis, L. tropica, L. major, species responsible for visceral and cutaneous leishmaniasis. No molecular methods and markers are currently validated to track drug-resistant organisms, and antimicrobial susceptibility tests are roughly not amenable to a clinical setting. Therefore, the delineation of drug-resistant Leishmania in the clinical setting is a prerequisite to applying system biology approaches to elucidate drug resistance. This vital point is, in my view, not discussed in this manuscript but is of uppermost importance to validate biomarkers identified in this study!!!!

Author Response

Thank you very much for your valuable comments. We completely agree with all comments and suggestions and made corresponding corrections in the manuscript. We have submitted the revised version of the manuscript with tracking.
